# Functional Neurorehabilitation in Dogs with an Incomplete Recovery 3 Months following Intervertebral Disc Surgery: A Case Series

**DOI:** 10.3390/ani11082442

**Published:** 2021-08-19

**Authors:** Ângela Martins, Débora Gouveia, Ana Cardoso, Carla Carvalho, Cátia Silva, Tiago Coelho, Óscar Gamboa, António Ferreira

**Affiliations:** 1Faculty of Veterinary Medicine, Lusófona University, Campo Grande, 1300-477 Lisboa, Portugal; 2Animal Rehabilitation Center, Arrábida Veterinary Hospital, Azeitão, 2925-583 Setúbal, Portugal; deborahisabel@msn.com (D.G.); anacardosocatarina@gmail.com (A.C.); mv.carla.c@gmail.com (C.C.); Catiamsilva@outlook.com (C.S.); tiagoccoelho@netcabo.pt (T.C.); 3CIISA—Centro Interdisciplinar-Investigação em Saúde Animal, Faculdade de Medicina Veterinária, Av. Universidade Técnica de Lisboa, 1300-477 Lisboa, Portugal; aferreira@fmv.ulisboa.pt; 4Superior School of Health, Protection and Animal Welfare, Polytechnic Institute of Lusophony, Campo Grande, 1300-477 Lisboa, Portugal; 5Faculty of Veterinary Medicine, University of Lisbon, 1300-477 Lisboa, Portugal; ogamboa@fmv.ulisboa.pt

**Keywords:** spinal cord injury, locomotor training, functional electrical stimulation, transcutaneous electrical SC stimulation, chronic dogs, 4-aminopyridine, neurorehabilitation

## Abstract

**Simple Summary:**

A non-invasive neurorehabilitation multimodal protocol (NRMP) may be applicable to chronic T3-L3 dogs 3 months after undergoing surgery for acute Intervertebral Disc Disease (IVDD) Hansen type I; this protocol has been shown to be safe, feasible, and potentially effective at improving ambulation in both open field score (OFS) 0 and OFS 1 dogs. The specific sample population criteria limit the number of dogs included, mainly due to owners withdrawing over time. Thus, the present case series study aimed to demonstrate that an NRMP could contribute to a functional treatment possibly based on synaptic and anatomic reorganization of the spinal cord.

**Abstract:**

This case series study aimed to evaluate the safety, feasibility, and positive outcome of the neurorehabilitation multimodal protocol (NRMP) in 16 chronic post-surgical IVDD Hansen type I dogs, with OFS 0/DPP− (*n* = 9) and OFS 1/DPP+ (*n* = 7). All were enrolled in the NRMP for a maximum of 90 days and were clinically discharged after achieving ambulation. The NRMP was based on locomotor training, functional electrical stimulation, transcutaneous electrical spinal cord stimulation, and 4-aminopyridine (4-AP) pharmacological management. In the Deep Pain Perception (DPP)+ dogs, 100% recovered ambulation within a mean period of 47 days, reaching OFS ≥11, which suggests that a longer period of time is needed for recovery. At follow-up, all dogs presented a positive evolution with voluntary micturition. Of the DPP− dogs admitted, all achieved a flexion/extension locomotor pattern within 30 days, and after starting the 4-AP, two dogs were discharged at outcome day 45, with 78% obtaining Spinal Reflex Locomotion (SRL) and automatic micturition within a mean period of 62 days. At follow-up, all dogs maintained their neurological status. After the NRMP, ambulatory status was achieved in 88% (14/16) of dogs, without concurrent events. Thus, an NRMP may be an important therapeutic option to reduce the need for euthanasia in the clinical setting.

## 1. Introduction

Spinal cord injury (SCI) is a commonly observed event in dogs following spinal trauma and Intervertebral Disc Disease (IVDD). Clinically, the SCI can be considered complete/incomplete if Deep Pain Perception (DPP) is absent (DPP−) or incomplete if DPP is present (DPP+) [1,2].

After IVDD Hansen type I, the spinal cord is able to sustain variable degrees of contusion and compression, depending on the extrusion velocity and amount of extruded disc material [3]. DPP− dogs can regain DPP after surgical decompression and thus return to a clinically ambulatory status [1,4,5]. Dogs without DPP recovery after surgery can become ambulatory through the reorganization of the spinal cord’s intrinsic circuit and central pattern generators (CPG) and with weight-bearing ability due to recovery of the residually intact descending axons traversing the lesion epicenter [6,7,8]. The absence of DPP is considered a negative prognostic factor for ambulation recovery [9]. However, if 10% of the axons in the malacic zone are preserved, there is a possibility of recovery [10].

More severely affected dogs can benefit from controlled activity programs in the field of neurorehabilitation. Functional neurorehabilitation belongs to physical medicine and rehabilitation in the area of restorative neurology and can be implemented in veterinary medicine [11], based on the evidence of signal transmission through the lesion, both caudal and rostral, as detected by electromyography for the motor tracts and evoked potentials for the sensory pathways [12,13,14].

Functional neurorehabilitation (FNR) protocols are based on repetitive task-specific therapy to optimize the spinal locomotor performance [15,16,17] and can stimulate and promote new connection formation, proving the spinal cord’s ability to interpret complex sensory inputs for posture and locomotion [8,18]. Therefore, FNR provides certain advantages in the recovery of muscle and motor functions in chronic dogs [19].

Dogs with chronic neurologic diseases can benefit from physical and rehabilitation support, which is common in human medicine, as well as in dogs and cats [11]. After SCI, both humans and cats can demonstrate spontaneous recovery, which can be explained by the similar cellular mechanisms responsible for spontaneous remodeling. Therefore, regarding the neurological clinical signs’ evolution, it is difficult to determine whether recovery is due to rehabilitation treatment or spontaneous recovery. Over time, in chronic dogs, it is possible that the neurological evolution could be mainly due to the neurorehabilitation protocol implemented [20,21,22].

Intensive FNR protocols modify motor interneuron intrinsic circuitry excitability and CPG through the stimulation of the cutaneous afferent pathway [23,24] and the amplitude of the motoneuron synaptic inputs, essentially from the axons descending in the ventrolateral funiculus (VLF) from the reticular formation [25], which involves the activation of the reticulospinal pathways [26,27,28,29]. Thus, increasing locomotor exercise intensity with repeated training can lead to significant changes in peak gait speed, stride length, and cadence [30]. Changes that promote the functionality of the neural network in animals with complete or incomplete SCI are entirely related to sensorimotor rehabilitation [7,31,32,33,34,35].

Quadrupedal step locomotor training could promote the activation of some long ascending and descending propriospinal interneurons [24,25] and improve the recruitment of muscles, mainly the flexors/extensors and adductors/abductors, which are very important for coordination and sensorimotor rehabilitation.

Pelvic limb muscle strength in SCI individuals can be improved by functional electrical stimulation (FES). FES includes efferent and afferent nerve stimulation, neuromuscular and muscle stimulation [12,36], and, depending on the cathode and anode orientation, has the potential to lead to regeneration and the activation of new connections [37,38].

Intralimb hip–knee coordination may be promoted by the combination of FES and locomotor training [37,38,39]. In chronic patients, electrical stimulation protocols with FES and Transcutaneous Electrical Spinal Cord Stimulation (TESCS) could be applied as a possible therapeutic neurorehabilitation tool for IVDD [40,41,42,43]. TESCS can lead to synergistic and interactive multi-segmental effects, converging with the ascending sensorial pathways and the descending motor pathways, that might be important in activating the spinal network [44]. Moreover, it is implied that this modality may stimulate new intraspinal circuits and persistent residual spinal circuits [12,45,46,47,48,49,50].

There have been investigations regarding pharmacological management with 4-AP in thoracolumbar chronic SCI dogs to improve neurological deficits [51,52,53]. The 4-AP is a potassium channel antagonist that could be used to block exposed fast-gated potassium channels in demyelinated axons and increase synaptic transmission at the pre-synaptic level [53]. Thus, it has been proposed as a possible intervention given its effect on the neural synaptic reorganization of the motor interneuron circuit and CPG—the spinal network.

Chronic dogs are usually not modulated considering peripheral spinal reflexes, manifesting a non-coordinated and non-synchronized flexion/extension locomotor pattern with clonic reflexes. It is believed that 4-AP administration may enhance motor neuron pool excitability in dogs with depressed activity; however, studies with larger cohorts are needed in order to determine how this drug could be useful [54].

Today, it is well known that the Central Nervous System (CNS) contains a complex network of neural connections, which are supported and modified by a population of glial cells, including astrocytes, oligodendrocytes, and microglia, which play an important role in neural plasticity. The astrocytes are the major components of the fibroglial scar that develops in the injury site after SCI [55]. This glial scar attempts to stabilize the injured parenchyma by reestablishing its physical and chemical integrity through different mechanisms, such as closing the blood–brain barrier, decreasing the infiltration of non-CNS cells, limiting possible infection [56], and playing an important pro-regenerative role [57].

Furthermore, the fibroglial scar, which contains a rich mixture of cells and extracellular matrix [58], also inhibits axonal growth and re-myelination.

Despite this disparity regarding the glial scar, there is increasing evidence that suggests that it can support CNS repair, essentially via astrocytes, NG2 glia, and microglia.

Historically, synaptic plasticity in pre-existent pathways and the formation of new circuits—in other words, synaptic and anatomical neuroplasticity, respectively—can promote the sprouting of lesioned fibers, contributing to regeneration [59]. Thus, the induction of an activity-dependent form of spinal cord plasticity may play an important role in the development of effective new rehabilitation modalities to be applied after SCI [60].

This case series clinical study intends to verify whether a neurorehabilitation multimodal protocol (NRMP) is a safe and feasible therapeutic protocol to be applied in SCI chronic dogs and to assess whether this NRMP allows a positive neurological evolution until ambulation and within a time period that is clinically acceptable.

## 2. Materials and Methods

This study was conducted between March 2011 and January 2021 at Arrábida Veterinary Hospital and Rehabilitation Centre, (HVA/CRAA) after obtaining approval from the Lisbon Veterinary Medicine Faculty Ethics Committee and after obtaining the owners’ consent.

### 2.1. Participants

Participants were included if they were chronic post-surgical IVDD Hansen type I dogs and more than 3 months had passed since their surgery. All had to be of chondrodystrophic breed, aged less than 7 years old, with a weight of less than 15 kg, with neuro-location at T10-L3, and classified at admission as paraplegic with DPP (OFS 1/ DPP+) or without DPP (OFS 0/DPP−). All DPP− dogs with voluntary movement of the tail, pain perception on the perineal region, scrotum, or vulva, and pain perception at the tip and base of the tail were excluded.

The dogs were evaluated and the findings were documented (Canon EOS Rebel T6 1300D camera) by two Certified Canine Rehabilitation Practitioners (CCRPs) from the University of Tennessee and a non-CCRP neurologist from Lisbon University.

The breed distribution was diverse: five (31%) Dachshunds, five (31%) French bulldogs, two (13%) Shi Tzus, two (13%) Pugs, one (6%) Pinscher, and one (6%) Pekingese. They had a mean age of 3.63 years (median of 4) and a mean weight of 7 kg (median of 7), and 56% (9/16) were males and 44% (7/16) were females. The most frequent neuro-location was L1–L2, for 37% (6/16) of dogs. Nine dogs were DPP− (56%) and seven dogs were DPP+ (44%).

### 2.2. Interventions

The dogs underwent a Home Work Plan (HWP) after the hemilaminectomy for Hansen type I IVDD, for 3 months, and performed three different types of protocols as described in Table 1, with no positive evolution for at least 6–8 weeks.

All were enrolled in the NRMP for a maximum of 90 days and were clinically discharged after achieving ambulation. Ambulation was considered when dogs showed “the ability to rise and take at least 10 consecutive weight-bearing steps unassisted without falling” [6]. Regarding DPP− dogs, they had to be able to perform curves without falling, with a wider support base and automatic micturition.

After admission, the dogs were redirected to a neurorehabilitation consultation and evaluated based on their history and neurorehabilitation examination (mental status, posture, OFS gait, postural reactions, peripheral spinal reflexes, cutaneous trunci muscle reflex, spinal cord palpation, and pain perception).

For DPP evaluation, 12 cm Halsted forceps were used to test the response to a noxious stimulus applied to the bone of both medial/lateral digits of the pelvic limbs, always in a controlled environment, regarding movement and noise. Moreover, to perform an accurate pain examination, we also tested the perineal region and the tip and base of the tail.

The flexion/extension locomotor pattern that was also monitored comprises interneurons interposed between the input and output. Thus, these interneurons allow divergence and promote input distribution to a wide population of output neurons. One type of input can then stimulate both agonist muscle contraction and antagonist muscle relaxation [61]. In SCI, a disruption in descending motor pathways, particularly in the lateral funiculus and the opposing pathways in the ventral funiculus, exerts a powerful excitatory influence on the alfa and gamma motoneurons of the extensor muscles [62]. A local spinal reflex and the upper motor neurons influence the activity of the alfa and gamma motoneurons, which allows the appearance of the crossed extensor reflex. Therefore, a normal dog, in the lateral recumbency position, should not exhibit the crossed extensor reflex, which is manifested as an ipsilateral flexion and contralateral extension reflex that helps to maintain posture when the dog withdraws one of its limps, such as in stepping.

#### 2.2.1. Neurorehabilitation Multimodal Protocol

The dogs were then subjected to an NRMP. For DPP+ dogs, the protocol comprised a combination of locomotor training and electrical stimulation. Electrical stimulation was based on Functional Electrical Stimulation (FES) and Transcutaneous Electrical Spinal Cord Stimulation (TESCS). For DPP− dogs, the protocol implemented was the same as described, with the inclusion of pharmacological management with 4-aminopyridine (4-AP).

This pharmacological management was applied in DPP− dogs because they were considered to have a complete SCI with a poor prognosis.

Locomotor training

The locomotor training began on the second day of admission after the dogs had become accustomed to the land treadmill. Dogs were adapted with the help of technicians in a calm environment.

The primary type of exercise was quadrupedal step training. For each training session, variables such as the walking speed and duration were increased and recorded, starting with 0.8 km/h (0.22 m/s) with a maximum of 1.9 km/h (0.53 m/s) [63,64,65] for 5 min (4–6 times/day, 6 days/week), to achieve 20 min (2–3 times/day, 6 days/week) [66]. The goal was to achieve 30–40 min (2–3 times/day, 5–6 days/week) [67], and the land treadmill slope was elevated to 10° [68] to a maximum of 25° [69], with forelimb–hindlimb coordination (Table 2).

The underwater treadmill (UWTM) training was introduced in all dogs from the second day of admission, with a water temperature of 26 °C [70] for 5 min to a maximum of 1 h training per day (5 days/week), monitoring for signs of overtraining. The speed ranged from 1 km/h (0.28 m/s) to 3.5 km/h (0.97 m/s) [26,71]. The training session consisted of increasing speed and duration under the same rule: four sessions with good performance indicated a 10% improvement in both variables [72,73,74] (Table 3).

Locomotor training was always associated with active and assisted kinesiotherapy exercises, such as postural standing, flexor movements, bicycle movements, balance board, and different floor surfaces [75] (Table 4).

Electrical Stimulation

The electrical stimulation protocol applied in all dogs was a combination of FES, aiming to achieve muscle contraction and neural connection, and TESCS to increase the descending pathway depolarization.

Functional Electrical Stimulation

A neuromodulation modality uses a short electrical pulse sequence to stimulate the lower motoneurons near the motor point region or by peripheric afferent stimulation, resulting in peripheral spinal reflexes [76,77]. This modality was performed in all dogs with superficial electrodes. One electrode was placed on the skin region corresponding to L7-S1 and the other electrode was close to the motor point of the hamstring muscles (biceps femoris, semitendinosus, and semimembranosus).

The current parameters were as follows: 60 Hz and 6–24 mA [78,79,80,81]; duty cycle: 1:4 ratio; ramp-up/down: between 2 and 4 s for ramp-up, 8 s on time, and 1–2 s for ramp down [82]. This was performed 2 to 3 times a day (5 days/week) and was discontinued as per neurological evolution (Figure 1).

Transcutaneous electrical spinal cord stimulation

This is a non-invasive and non-painful rehabilitation therapeutic modality with the ability to neuromodulate the spinal cord’s physiological status [83].

All dogs underwent TESCS three times a day (5 days/week), which was gradually discontinued when the flexion/extension locomotor pattern appeared; thus, the surface electrodes were placed in the paravertebral muscles (one electrode on L1-L2 and the other on L7-S1) and a continuous current of 50 Hz and 2 mA was applied for 10 min (Figure 2A,B).

Pharmacological management

After 4 weeks of the NRMP, dogs who displayed a flexion/extension locomotor pattern but were DPP− started pharmacological management in combination with the training protocol, after the owners’ consent was obtained.

This pharmacological treatment was based on a potassium channel blocker, 4-AP [53,84,85,86], according to the following protocol: 0.3 mg/kg per os BID 3 days; 0.5 mg/kg BID 3 days; 0.7 mg/kg BID 3 days; 1.1 mg/kg BID for 21 days. In the event of side effects (e.g., vomiting, diarrhea, and seizures), dogs would have been immediately treated and withdrawn from the study; however, none of the dogs showed adverse effects in response to this medication that warranted withdrawal.

Supportive care

All the dogs had neurogenic bladders; therefore, their bladders were manually expressed 3–4 times/day. Urine was monitored daily for odor and color changes, and in the case of suspected urinary tract infection (UTI), urine culture (by cystocentesis) and specific antibiotic treatment were performed. Only two dogs were UTI-positive and required antibiotic therapy with water consumption control. The dogs were maintained on a full-time hospitalization regimen and rested in soft beds. Feeding was performed three times/day (intake increase of 30%) and a hydric support of 100–120 mL/kg/day was provided after resistance training (training intended not to reach the maximum activity within an extended time period) that was alternated with fortification training (training intended to achieve the maximum activity within a short time period) according to the dog’s needs. All were trained during the day, starting at 9:00 a.m. and finishing at 7:00 p.m., assisted only by veterinarians and veterinary nurses that were CCRPs.

#### 2.2.2. Outcomes/Follow-Up

During full-time NRMP, the dogs were evaluated neurologically every 3 to 7 days in order to make necessary changes concerning exercise duration and intensity. This evaluation included the assessment of the flexor reflex, flexion/extension locomotor pattern, postural standing, and the OFS [87], including deep pain assessment. Thus, pain perception assessment was specifically evaluated in all dogs as described above.

Outcomes were performed at 15, 30, 45, 60, 75, and 90 days, and they were followed up after 8–10 days, 1 month, and 6 months by the same CCRP instructor/examiner, based on the neurorehabilitation examination criteria and OFS.

The primary outcome parameters of this study were the ambulatory status with an OFS ≥11, which indicates ambulation for DPP+ dogs. Regarding DPP−, ambulatory dogs will show spinal reflex locomotion (SRL). DPP− dogs without ambulatory status will show no spinal reflex locomotion (NSRL). Dogs with NSRL demonstrated the flexion/extension pattern but without the ability to rise.

A secondary outcome was the pain perception assessment, evaluated on the tip and base of the tail and the perineum region (S1 and S2 dermatomes). Deep Pain Perception was assessed with 12 cm Halsted forceps in the lateral and medial digits of both pelvic limbs in a controlled environment regarding noise and the circulation of people.

The evolution of the 16 chronic dogs was recorded after HWP and NRMP implementation, as illustrated in the flow diagram, as well as at follow-up. All the dogs were medically discharged when they achieved ambulation, with subsequent assessment by long-term follow-up after 8–10 days, 1 month, and 6 months (Figure 3).

#### 2.2.3. Statistical Analysis

Database and statistical analysis were performed using Microsoft Office Excel 2016^®^ (Microsoft, Redmond, WA, USA). Categorical data were described as frequencies and proportions (with 95% confidence intervals). The collected data included breed, sex, age, weight, lesion neuro-location, DPP presence, OFS at admission and at each time point (15, 30, 45, 60, 75, and 90 days), ambulatory status, time when ambulation was achieved, time of medical discharge, follow-ups (8–10 days, 1 month, and 6 months), and the presence of concurrent events. Concurrent events during the NRMP included: pressure ulcers, alveolo-interstitial pulmonary disease, gastrointestinal disease, and bacteria/fungal dermatitis.

## 3. Results

The NRMP was applied to nine DPP− dogs (OFS 0) and seven DPP+ dogs (OFS 1) for a maximum period of 90 days. All the dogs had been paraplegic for more than 3 months after IVDD surgery and HWP. Dogs who achieved ambulation were discharged, as demonstrated in Figure 4.

As mentioned above, data were collected regarding breed, age, gender, weight, neuro-location, and the presence of DPP. Moreover, this study included 100% (*n* = 16) paraplegic dogs, of which 56% (9/16) were DPP− and 44% (7/16) were DPP+.

At medical discharge and after the NRMP, ambulatory status was achieved in 88% (14/16) of dogs. No concurrent events were observed during the NRMP.

### 3.1. DPP− Dogs’ Results

Among the nine DPP− dogs that were admitted with an OFS of 0, 78% (7/9) were discharged with SRL and 22% (2/9) remained with NSRL.

On day 15 and day 30, the dogs showed no significant evolution. Adverse events, such as vomiting, diarrhea, and seizures, were not seen in any of the dogs receiving 4-AP medication.

At outcome day 45, two dogs (22% (2/9)) obtained SRL with medical discharge at day 60. On day 60, two more dogs (22% (2/9)) obtained SRL with medical discharge at day 90. Three more dogs (33% (3/9)) achieved SRL at outcome day 75, remaining until day 90. Only two dogs (22% (2/9)) remained with NSRL on day 90. The appearance of SLR occurred only after day 30, which may have been coincidental or may have been due to the contribution of 4-AP to their recovery.

At the 8–10-day follow-up, all nine dogs maintained similar neurological status, as well as at the 1-month follow-up, although one dog missed their follow-up appointment. At the 6-month follow-up, only six dogs attended, and they all maintained the same neurological status. The evolution of the ambulatory status and medical discharge statistics are presented in Figure 5, with the OFS 0 dogs represented in blue.

The mean time in which ambulation was achieved, in DPP− dogs, was 62 days (median 60 days) and all regained automatic micturition.

### 3.2. DPP+ Dogs’ Results

Seven DPP+ dogs were admitted with an OFS of 1, and 100% were discharged with ambulatory status.

On outcome day 15, all dogs showed an improvement in the OFS (range 3 to 9) and two dogs achieved ambulatory status at outcome day 30: one with OFS 13 and the other with OFS 11. The first one was medically discharged on day 30, and the second only on day 60.

On outcome day 45 (*n* = 6), two more dogs achieved ambulatory status with an OFS of 11, and on outcome day 60 (*n* = 6), all the dogs were ambulatory, with two medical discharges with an OFS of 13.

On outcome day 75 (*n* = 4), three dogs had OFS 12 and one had OFS 11. On day 90, the former improved to OFS 13 and the latter to OFS 12. The OFS results were registered during the outcomes, as shown in Table 5, and their evolution is displayed in Figure 6.

The mean time to achieve ambulation in DPP+ dogs was 47 days (median 45 days) and all achieved voluntary micturition.

The evolution of the ambulatory status and medical discharges are presented in Figure 5, with the OFS 1 dogs represented in brown.

At the 8–10-day follow-up, all seven dogs maintained similar OFS results, and at the 1-month follow-up, two dogs improved to an OFS of 14. At the 6-month follow-up, one dog did not attend, two improved to an OFS of 14, and one improved to an OFS of 13. The evolution of mean OFS results since medical discharge and during follow-up is represented in Figure 7.

## 4. Discussion

In this study’s population, the median age was 4 years old and 31% (5/16) of the dogs were French Bulldogs, which can be explained by the fact that this breed is fixed for the CFA12 FGF4 retrogene at a significantly younger age, as reported by Dickinson et al. (2020) [88], with a median age of 3.7 years. Dachshunds displayed the same prevalence (5/16); this breed is homozygous for the same retrogene but with a median age of 6.5 years [89], also according to our population inclusion criteria. The presence of the CFA12 FGF4 retrogene seems to be sufficient to cause the premature degeneration of the intervertebral disc [89], associated with early chondroid metaplasia in chondrodystrophic breeds [90], which includes all our dogs.

Regarding this study’s population, the mean weight was 7 kg, in line with most recent studies [91,92,93]. The same comparison could be made regarding the lesion site, focusing on the thoracolumbar region, which can be justified by anatomical reasons [2,92,93].

In the present study, these dogs had a chronic spinal injury that was present more than 3 months after surgery. Thus, the positive evolution in the locomotor pattern to achieve ambulation may indicate the benefits of the NRMP [16]. This type of neurorehabilitation protocol originated in human medicine; however, there are also several recent studies that use a population of chronically injured dogs that simulate a situation in humans [94,95,96,97,98].

The NRMP implemented was based on locomotor training, which included land treadmill by body weight supported treadmill training (BWSTT), UWTM, and kinesiotherapy exercises [99], as recently reinforced by the Canine Spinal Cord Injury Consortium (CANSORT SCI) [6].

The land BWSTT enhances muscle endurance, promotes the rhythmic activity of the spinal network, and increases the stimulation of the remaining descending/ascending pathways [17,100]. Locomotor training is essential as a strong neuromodulator that may contribute to neuroregeneration and to the disinhibition of the descending motor pathways that are inhibited in SCI dogs, stimulating pre-motor neuronal control and promoting intra-limb and inter-limb coordination [101,102,103].

In this case series study, the results showed that in DPP– dogs, the neurological deficits improved after implementation of the pharmacological approach, with one dog achieving SRL 15 days after starting 4-AP, i.e., at the 45-day outcome. Moreover, 78% of ambulatory DPP− dogs achieved SRL by the 75-day outcome, i.e., within two and a half months. Thus, the mean time to achieve ambulation in DPP− dogs was 62 days and all achieved automatic micturition. At follow-up, all dogs maintained similar neurological status, due to the spinal cord’s memorization of the locomotor pattern. These results suggest that more studies are needed in this field.

Furthermore, it is important to mention that, to the best of our knowledge, there is no study that unites the effects of intensive locomotor training, electrical stimulation, and 4-AP administration. Therefore, this is the first case series study that may prove the relevance of a multimodal approach.

Lewis and colleagues (2019) [54] administered a mean drug dosage of 0.78 mg/kg, with all the dogs receiving at least three 4-AP doses, separated by at least 8 h. In the present study, the dosage was gradually administered from 0.3 mg/kg until 1.1 mg/kg per os BID, to avoid increased drug-related side effects, such as seizure activity, as reported in the aforementioned study. There were no adverse effects reported in our results, which suggests that this was a safe and non-harmful protocol.

In order to be able to repeat this NRMP, it is necessary to use an electrical stimulation device, a land treadmill, and, ideally, a UWTM (given the possibility to stimulate afferent pathways, cutaneous receptors, and proprioceptive receptors) [104,105,106]. Moreover, given the benefits of water, such as floatability, viscosity, and resistance, superficial tension and vasodilation/vasoconstriction are promoted by the water temperature [107,108,109,110].

In our dogs, pharmacological management was only applied within a period of two months. The possible lack of response may suggest that there is a limited functional effect of 4-AP in chronic SCI dogs when the neuronal matter is already in a state of hyperexcitability [111,112]. However, it is also suggested that 4-AP can potentiate the spinal network in chronic dogs, with the observation of low spasticity signs [54], important given the fact that severe chronic thoracolumbar SCI dogs usually develop signs of spasticity in weeks to months after the initial injury [113,114]. Thus, 4-AP could possibly decrease clonic reflexes and increase a synchronized and coordinated flexion/extension locomotor pattern.

In chronic SCI, the axons fail to regrow, and this axonal regrowth could be inhibited by scar tissue formation [115]. The glial scar and cavity formation are highlighted as the main pathophysiological feature in chronic SCI, characterized by reactive astrocytes in the chronic phase after the injury [116]. When mainly accentuated within the grey matter, it could be related to severe necrosis, with the destruction of the tissue structure [117]. At the same time, extensive oligodendrocyte remyelination of the remaining axons could also occur [116].

A recent analysis that assessed transcranial magnetic motor evoked potential suggested that spinal locomotion could be associated with remaining intact conduction through the descending motor tracts [5].

Granger et al. (2012) [94] and Hu et al. (2018) [118] studied the hypotheses regarding the relation between spinal walking and spinal “long tract” conduction. It was demonstrated that in some dogs, even those with complete SCI, a persistent passage of electrophysiological stimuli across the lesion with recorded spinal somatosensory evoked potentials above the injury [118], which could also be seen in humans [119]. In the aforementioned study, it was suggested that this remaining passage is a possibility, or not, in spinal walking dogs; however, Lewis et al. (2017) [5] support this positive relationship and mention a non-relation between the lesion site and recovery of spinal walking.

Hu et al. (2018) [118] concluded that, in chronic dogs, this remaining electrophysiological conduction throughout the long tracts is extremely variable and there is still very little evidence regarding the benefits in improving functional outcomes. Other studies in humans have suggested that propriospinal connections could bypass the injury site, improving functional recovery [120,121,122,123,124,125,126].

Locomotor training activity after SCI is a well-established modality to be applied after SCI, promoting the recovery of stepping in cats [124,125]. There is also evidence that the same could be seen in clinically spinal-cord-injured dogs, as shown by Gallucci et al. (2017) [92], but also in IVDD and traumatic SCI dogs.

All DPP− dogs in this study manifested some spinal network plasticity, allowing a flexion/extension locomotor pattern with a fast but modulated flexor reflex and crossed extensor reflex, characteristic of the NSRL [126,127], which was obtained in 22% (2/9) through the quadrupedal training with sensory feedback, which sends signals to the CPG [28,83,123,128]. The remaining seven dogs achieved ambulation by SRL, through the possibility of SCI regeneration and/or the activation of residual descending pathways [5].

In the chronic phase, mainly in DPP− dogs, there is no standard treatment to regain ambulation, aiming to promote synapse plasticity, increase axonal sprouting, and lead to regeneration [129]. The existing gap in chronic post-surgical IVDD dogs has led to recent research in regenerative medicine that has investigated Schwann cells [130], embryonic stem cells [131], macrophages [132], neural stem cells [133], and pluripotent stem cells [115,134]. All these studies aimed to demonstrate the effects of regenerative medicine on axonal regeneration with possible benefits in recovery.

It was shown that neurogenic induction, in both canines and humans, with dental pulp stem cells (neuroectodermal stem progenitor cells—NSPCs) could cause morphological changes [135].

In rats with spinal cord transection, the combination of NSPCs with electroacupuncture has been shown to improve neurological evolution and regeneration [136,137]. Similar studies with the association between locomotor training and stem cells were performed [97].

In Prado et al. (2019) [138], in 16 dogs ranging from 5 to 11 years old, with acute IVDD, who remained paralyzed for at least 3 months and were subjected to dorsal hemilaminectomy, stem cells were injected intra-medullary into the spinal cord parenchyma—in the middle, cranial, and caudal to the lesion. A second transplantation was performed after seven days, followed by three percutaneous intraspinal injections. Locomotor training 3 times a week was performed after 8–11 days, for the initial 7 weeks, followed by twice a week for a further 5 weeks, in association with UWTM twice a week, for 3–15 min, for a total of 12 weeks. The results demonstrated that no beneficial effects could be associated with the cells alone or with the combined treatment.

Compared to the present study, the results described above may be related to the fact that locomotor training should be performed every day in order to induce plasticity in the reflex pathways [17,126,139,140]. In Martins et al. (2020) [17], the authors reported a statistically significant positive evolution over time, with an increase in mean OFS related to the locomotor training implemented. Thus, repetitive locomotor training tasks could promote cutaneous reflex excitability [141], by different mechanisms, such as the sprouting and regeneration of damaged fibers and changes in synapses’ efficiency [142,143].

One other study that used adipose tissue mesenchyme stem cells (AT-MSCs) demonstrated a regenerative function, either by mediating replaced lost cells, such as oligodendrocytes, which would facilitate the injured axons’ remyelination, or by mediating astrocytes. Therefore, some degree of neuroplasticity was evolved in axonal sprouting and the formation of new circuits. Moreover, they reported that the locomotor recovery was obtained within 8 weeks, with a tendency to stabilize between 8 to 16 weeks [129].

In the present study, 100% of the DPP+ dogs achieved ambulatory status at clinical discharge. On day 15, all dogs had improved but only two achieved ambulation by day 30, the time point at which an OFS peak is achieved, as seen in Figure 6. At outcome day 45, the mean OFS was 10.7, and at day 60, it increased to 11.8, as shown in Table 5. Such results may suggest that chronic post-surgical dogs need more time to achieve ambulation, within a mean time of 47 days (median 45 days). This is in contrast to acute post-surgical IVDD dogs, who usually demonstrate 90% to 100% of recovery, with 70.8% recovering within the first 15 days [144]. All DPP+ dogs achieved voluntary micturition. At follow-up, the average evolution was positive regarding the neurological status (Figure 7).

Regarding DPP− dogs, there was no DPP recovery, in contrast to the acute dogs, who usually regain 41–62% of ambulation [1,145,146,147,148,149]. According to the Canine Spinal Cord Injury Consortium (CANSORT SCI), DPP– dogs recover DPP to ambulation ~60% within 6 months after injury. In our results, DPP– chronic dogs achieved 78% of ambulation within 2 and a half months; however, all had achieved the flexion/extension locomotor pattern within the first two weeks, which allowed the introduction of the pharmacological management protocol with 4-AP.

Since this was a case series study, we suggest that more studies are needed in this area, given the clinical interest and the need to be open to multidisciplinary approaches, mainly on post-surgical DPP− chronic dogs. This study’s population was difficult to obtain due to the strict inclusion criteria, because chronic post-surgical IVDD Hansen type I paraplegic dogs do not typically survive. We propose a multimodal approach in order to achieve ambulation, decrease the need to perform euthanasia, and promote the quality of life of these dogs.

Thus, limitations included the small number of dogs, due to the criteria mentioned above, and the lack of control cases, which is a significant limitation of this study. The appearance of the SRL may have been due to spontaneous recovery, coincidental, or due to the contribution of 4-AP, and a control group would be required in order to assess this.

Future controlled studies with a larger number of dogs with chronic DPP+ or DPP− incomplete recovery are needed. It would also be interesting to continue this study considering the association between the NRMP, stem cell administration, and/or a pharmacological management protocol with 4-AP.

## 5. Conclusions

This prospective case series clinical study proposes a neurorehabilitation multimodal protocol (NRMP) as a safe and feasible therapeutic approach that was applied in all 16 chronic post-surgical SCI dogs without any side effects.

Chronic dogs with OFS 0/DPP− (*n* = 9) could not recover DPP within the 90 days of the NRMP, but 78% (7/9) were able to achieve ambulation by SRL, with a mean time of 62 days. Thus, it is suggested that this type of ambulation may be possible with a multidisciplinary protocol with the influence or not of spontaneous recovery. Moreover, it is possible that dogs that were admitted in a more severe state (DPP−) may simply take longer to respond to the non-pharmacological rehabilitation modalities when compared to DPP+ dogs.

On the other hand, dogs with OFS 1/DPP+ (*n* = 7) were able to achieve 100% of ambulation within a mean period of 47 days, suggesting the need for a longer period of time when compared to the average acute dogs who are submitted to similar rehabilitation approaches.

The type of population considered in this study, which achieved 88% (14/16) of ambulation, usually could not persevere. Thus, an NRMP may be an important therapeutic strategy to reduce the need for euthanasia in the clinical setting. This study should be continued, given all the limitations mentioned.

## Figures and Tables

**Figure 1 animals-11-02442-f001:**
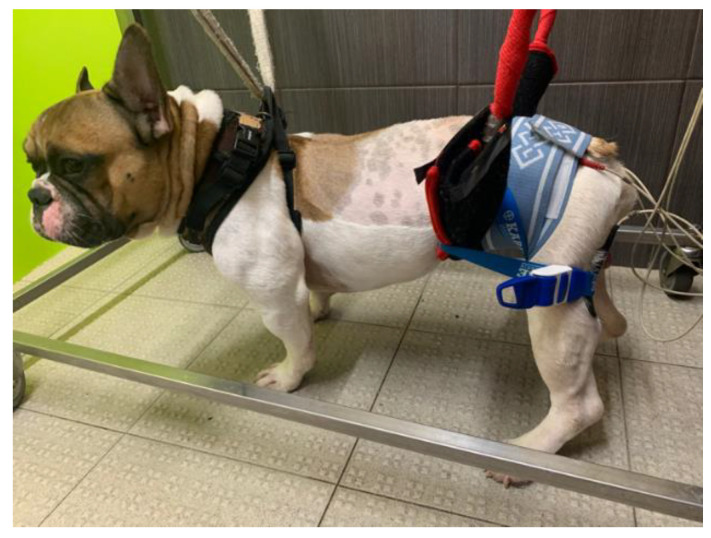
Functional electrical stimulation. One electrode was placed at L7-S1 region and one close to the motor point of the hamstring muscles (biceps femoris, semitendinosus, and semimembranosus).

**Figure 2 animals-11-02442-f002:**
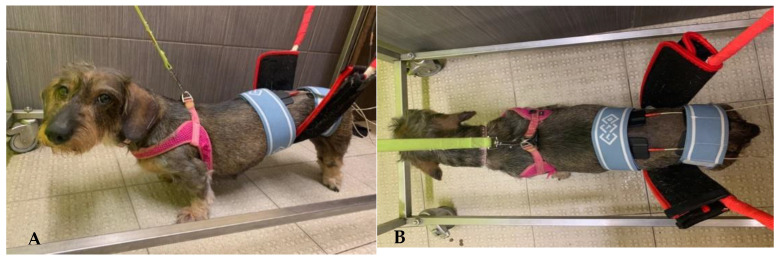
(**A**,**B**) Transcutaneous electrical spinal cord stimulation. One electrode placed on L1-L2 and the other one on L7-S1.

**Figure 3 animals-11-02442-f003:**
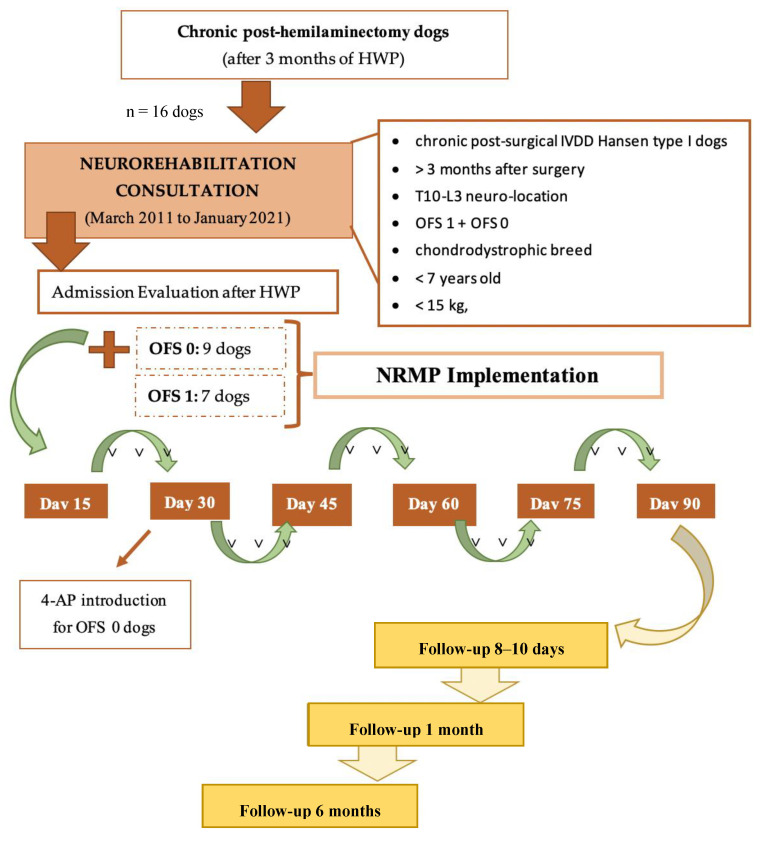
Flow diagram illustrating the study design of chronic paraplegic post-hemilaminectomy dogs. Abbreviations: OFS—open field score [64]; NRMP—neurorehabilitation multimodal protocol; HWP—Home Work Plan; SRL—Spinal Reflex Locomotion; NSRL—Non-Spinal Reflex Locomotion.

**Figure 4 animals-11-02442-f004:**
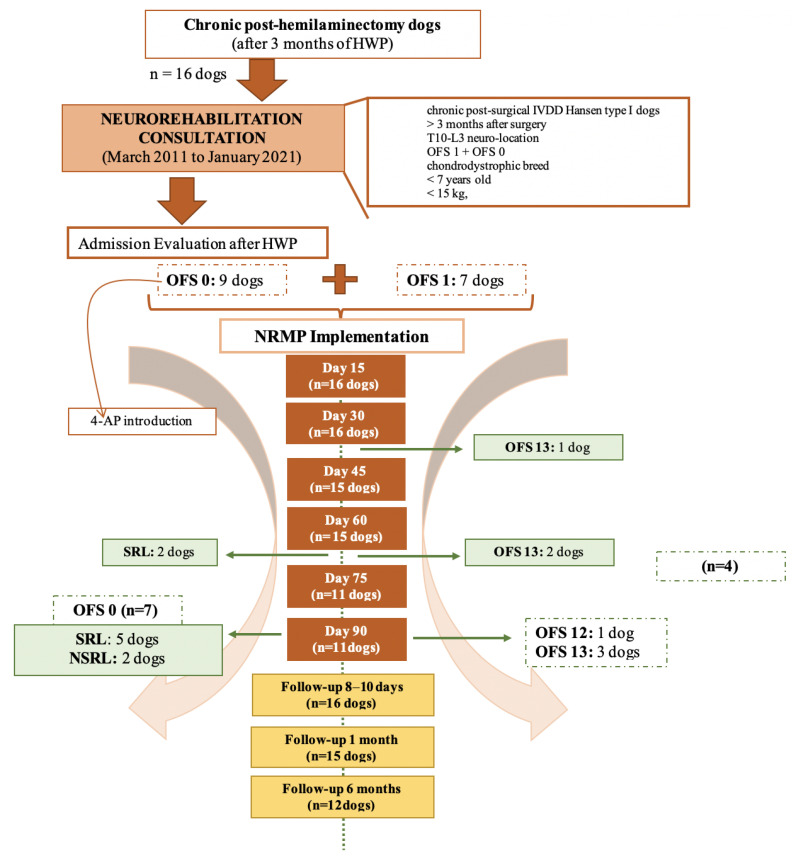
Flow diagram illustrating chronic paraplegic post-hemilaminectomy dogs’ evolution and medical discharge. Abbreviations: OFS—open field score [64]; NRMP—neurorehabilitation multimodal protocol; HWP—Home Work Plan; SRL—Spinal Reflex Locomotion; NSRL—Non-Spinal Reflex Locomotion.

**Figure 5 animals-11-02442-f005:**
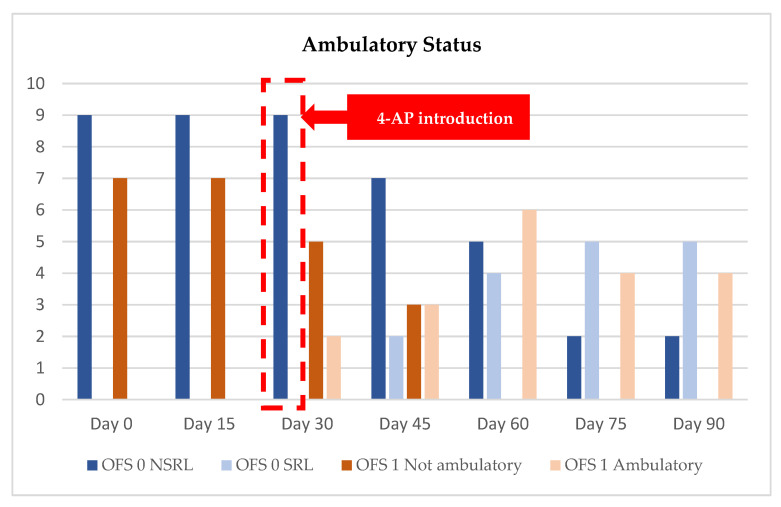
Evolution of ambulatory status and medical discharge from admission (day 0) until day 90. OFS—open field score [64]; SRL—Spinal Reflex Locomotion; NSRL—Non-Spinal Reflex Locomotion; OFS 0: blue color; OFS 1: brown color.

**Figure 6 animals-11-02442-f006:**
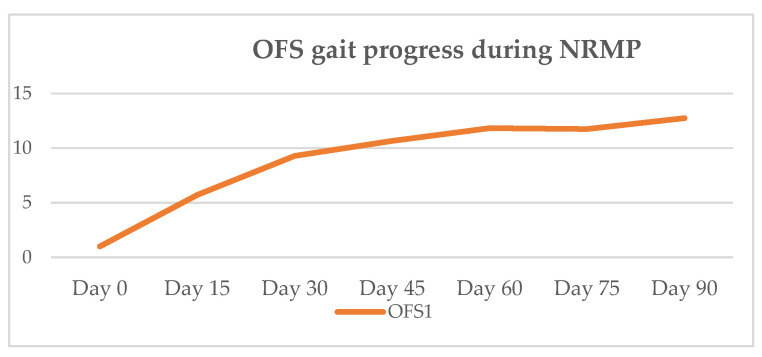
Mean OFS gait progression of OFS 1 dogs during NRMP. OFS—open field score [64]; NRMP—neurorehabilitation multimodal protocol.

**Figure 7 animals-11-02442-f007:**
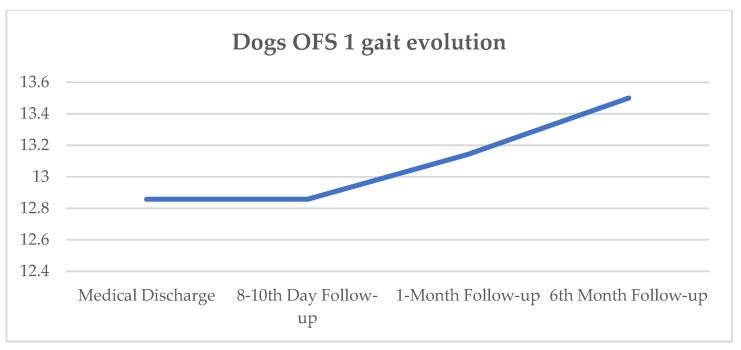
Dogs’ OFS gait evolution since medical discharge and follow-up (8–10-day, 1-month, and 6-month); OFS—open field score described by the y-axis [64].

**Table 1 animals-11-02442-t001:** Home Work Plan: Different types of protocols.

Protocol	Description	Frequency
A	30 min land treadmill	3 days/week
B	1 h leash ground walk	4 days/week
C	30 min underwater treadmill	1–2 days/week

**Table 2 animals-11-02442-t002:** Land treadmill locomotor training.

Week	Time Duration	Walking Speed	Frequency	Slope
1st–2nd	Up to 5–10 min	0.8–1.9 km/h	4–6 times/day 6 days/week	-
3rd–4th	Up to 20 min	2 km/h	2–4 times/day 6 days/week	-
5th–6th	Up to 30 min	2.2 km/h	2–3 times/day 6 days/week	-
7th–8th	Up to 40 min	2.5 km/h	2 times/day 6 days/week	5°
9th–10th	Up to 40 min	2.5 km/h	2 times/day 5 days/week	10°
11th–12th	Up to 40 min	2.5 km/h	1 time/day 5 days/week	25°

**Table 3 animals-11-02442-t003:** Underwater treadmill locomotor training.

Week	Time Duration	Walking Speed	Slope
1st–2nd	Up to 5–10 min	1–1.2 km/h	-
3rd–4th	Up to 10–20 min	1.8–2 km/h	-
5th–6th	Up to 30 min	2–2.5 km/h	5°
7th–8th	Up to 40 min	2.8–3 km/h	5°
9th–10th	Up to 40 min	3–3.5 km/h	5°
11th–12th	Up to 60 min	3.5 km/h	10°

Frequency of underwater treadmill training: 5 days a week, once a day, always in the morning.

**Table 4 animals-11-02442-t004:** Kinesiotherapy exercise training.

Week		Exercise	Time Duration	Frequency
1st–4th		Postural standing	Until 10 min	2–4 times/day 6 days/week
		Flexor movements	10–20 repetitions	2–3 times/day 5 days/week
		Bicycle movements	5–10 repetitions	
5th–8th	Dog with active postural standing	Different floors gait stimulation	3–5 min	2–3 times/day 5 days/week
	Dog without active postural standing	Postural standing	10 min	
		Different floors gait stimulation	3–5 min	
9th–12th		Different floors gait stimulation	5–10 min	2 times/day 5 days/week
		Balance board	5 min	

All exercises were performed accordingly to dog’s cardiorespiratory ability.

**Table 5 animals-11-02442-t005:** OFS gait data records and mean time since admission and during outcomes (day 15, 30, 45, 60, 75, and 90), OFS at medical discharge and during follow-up (8–10-day, 1-month, and 6-month).

DPP+ Dogs ID	Day 0	Day 15	Day 30	Day 45	Day 60	Day 75	Day 90	OFS Medical Discharge	8–10-Day Follow-Up	1-Month Follow-Up	6-Month Follow-Up
**1**	1	3	7	10	11	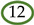	13	13	13	13	14
**2**	1	4	9	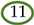	13			13	13	14	14
**3**	1	3	7	10	11	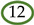	13	13	13	13	14
**4**	1	7	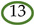					13	13	13	13
**5**	1	7	9	11	12	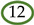	13	13	13	13	13
**6**	1	9	11	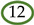	13			13	13	14	
**7**	1	7	9	10	11	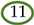	12	12	12	12	13
**OFS Mean**	1	5.7	9.3	10.7	11.8	11.8	12.8				

## Data Availability

The data presented in this study are available upon request from the corresponding author.

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
