# Peer review of "Functional Neurorehabilitation in Dogs with an Incomplete Recovery 3 Months following Intervertebral Disc Surgery: A Case Series"

_animals, 2021, doi:10.3390/ani11082442_

Round 1
Reviewer 1 Report
This study is well done and the data from a long period is difficult to collate, so congratulations for doing so. There is a fair amount of English language grammar editing that is required to make this paper of a scientifically accepted caliber.
A few additional recommendations:
-Many of the ideas you outline in the discussion section should be brought to the introduction. A stronger introduction will allow a better/stronger overall paper. Include ideas about the use of 4-AP as well as the use of neurorehabilitation to improve neuroplasticity in patients with ongoing glial scarring and reduced neuro-responsiveness.
-Additional definitions should be added to methods and materials. E.g. defining flexion/extension reflexes, the clinical significance of crossed-extensor reflexes and the definition of a "concurrent event"
-Tables and figures are helpful and professional but need some cleaning up. Tables should be numbered separately from figures. The y-axis of tables are not labeled and should be labeled.
-Unclear why tail and perineum pain perception testing were excluded in patient selection but then became a major part of the discussion. Just clarify more thoroughly.
-In line #330, indicate that the therapy exercises chosen were similar to those reported by Lewis, but not as reported by Lewis since the paper cited was published long after the study period and data collection began.
-Line #376- no adverse events instead of no secondary events makes it more clear that vomiting, diarrhea and seizure was not seen in patients receiving meds in the study. This should also be reported in the results section.
-The paragraph beginning at line # 461 is out of context. Clarify from the beginning of the paragraph how these patients were treated and why they didn't recover.
Reviewer 2 Report
This is an interesting study on use of neurorehabilitation (NMRP) in chondrodysplasitc dogs with paraplegia due to intravertebral disc disease.
There is a significant shortcoming of study design in that the authors did not compare the treated dogs to a control group (dogs with the same condition that did not undergo NMRP. Thus, no conclusions can be drawn regarding the contribution of NMRP to outcome. I strongly recommend that the authors add a control group to the study, possibly retrospectively from records of dogs that did not receive this same intensive NMRP and either underwent no rehabilitation, or a less intensive but standard protocol. if this is not possible, the authors should at least address the lack of control cases as a significant limitation to their study.
In the conclusions section, the authors state that ambulation in DPP- dogs seemed only to be possible after starting pharmacological management (with 4-AP) at day 30. However, since there was no control group within this sub-group, the later onset of SLR ambulation in DPP- dogs may have been coincidental, and not actually related to starting the 4-AP treatment.
Why were the DPP- dogs not started on 4-AP at the same time as the DPP+ dogs? Understanding that there might be a medical reason for this, but the authors should explain this in the materials and methods section. As it is now, it appears to be evaluating the effectiveness of 4-AP in dogs with IVDD/SCI, but comparing dogs of different levels of severity, and I do not believe that was the intention of the study.
Line 106-107: The authors note that the most frequent location for lesion was L1-L2 with 37% in that location. Where were the other locations of lesions? This specifically becomes of interest on lines 167 (FES) and 182 (TESCS) where the authors describe location of the electrodes at L7-S1 and L1-L2/L7-S1 respectively, as the inclusion criteria stated lesions had to be between T10-L3.
Lines 193-194 describes reasons a dog may be withdrawn from the study. Did this occur in any of the dogs? i.e. Was the starting number of dogs 16 and none were withdrawn or was the starting number higher and some were withdrawn? Since the texts states "...dogs were immediately treated and withdrawn from the study." it implies that it did occur. If none of the dogs whad to be withdrawn, suggest changing the text to read "...In case of side effects (e.g. vomiting, diarrhoea, and seizures, dogs would have been immediately treated and withdrawn from the study, however, none of the dogs showed adverse effects of this medication warranting withdrawal.
Lines 204-204 mention resistance training and fortification training without further description. Please briefly describe what this entailed as the terminology might not familiar to many readers.
Line 262: This sentence may not be grammatically correct and is awkward to read. "Only after pharmacological approach with 4-AP, was seen SLR." However, this line is also misleading as it implies that the appearance of SLR was the result of starting 4-AP, which you cannot conclude as there was no control group not receiving the 4-AP. Suggest rewriting to specify that appearance of SLR may have been coincidental and that assessment of the contribution of 4-P to this recovery would require a control group in future studies.
Line 268: This sentence ends awkwardly "...although one dog missed." Suggest rewriting to read "...although one dog missed their follow-up appointment"
Line 497-499: Again, cause and effect relationship between 4-AP and recovery to SLR cannot be concluded without a control group. "This ambulation seemed only possible after starting pharmacological management with 4-AP." It also also seems possible that dogs that started the study in a more severe state (DPP-) may simply take longer to respond to the non-pharmacological rehabilitation modalities than DPP+ dogs.
